# Towards Efficient Confidence Estimation for Large Language Model Reasoning

## Abstract

Recent advances have demonstrated the powerful reasoning capabilities of large language models (LLMs), and accurately measuring the confidence of reasoning paths is crucial for improving the performance and trustworthy of AI systems. Benefiting from consistency function for reasoning, the self-consistency method often provides an effective confidence estimation. However, it suffers from the variance issue, which extremely constrains the performance when the sampling is insufficient. Existing methods such as the temperature sampling cannot well resolve this problem as it not only necessitates a calibration set but also tends to sacrifice the reasoning capability of LLMs. In this paper, we propose a data-free, and highly sampling efficient method to control the variance. The merit of our approach lies in a reasonable integration of the LLM's probability estimation and the self-consistency confidence. Our theoretical analysis confirms the efficacy of our method by achieving a lower estimation error and a higher error reduction rate. Furthermore, an in-depth analysis of the error decomposition reveals an improved technique, which can significantly improve error reduction rate with only a small scale of bias induced. Experimental results across seven benchmark datasets demonstrate that our proposed approaches achieve superior confidence estimation, boosting the accuracy on both mathematical reasoning tasks and code generation tasks. Our code is provided in the supplementary material.

## 1 Introduction

Recently, large language models (LLMs) have made significant progress in various applications, including problem solving (Lewkowycz et al., 2022b; Li et al., 2024), planning (Valmeekam et al., 2023; Deng et al., 2024), and decision making (Ouyang & Li, 2023; Sblendorio et al., 2024), showcasing their strong reasoning capabilities. The confidence in reasoning, i.e., the likelihood of the reasoning answer being correct, can help refine the answers from reasoning paths (Wang et al., 2022), enhance the interpretability of reasoning results (Stengel-Eskin & Durme, 2023), and ultimately contribute to building trustworthy artificial intelligence systems (Guo et al., 2017; Felicioni et al., 2024). However, recent studies (Shen et al., 2024; Geng et al., 2024; Zhao et al., 2024) have shown that LLMs often fail to provide reliable confidence estimates, underscoring the importance of accurate confidence estimation for LLM reasoning.

There are several typical confidence estimation methods for LLM reasoning, i.e., perplexity (Chen et al., 1998), verbalized confidence (Xiong et al., 2023; Tian et al., 2023), and self-consistency confidence (Wang et al., 2022; Chen et al., 2023). Among them, our empirical studies show that, on math reasoning tasks, self-consistency confidence with a proper consistency function can consistently provide superior and satisfactory performance in both accuracy and calibration error metrics. However, the self-consistency confidence suffers from the large variance issue when the sampling is insufficient. This is because an accurate self-consistency confidence is computationally infeasible, and existing methods tend to adopt the Monte-Carlo sampling as an estimation, which suffer from low variance reduction efficiency of a linear rate.

A straightforward method to reduce the variance could be controlling the sampling temperature, since the decrease of temperature can narrow down the sampling space and alleviate the requirement of sample size. Nevertheless, the tuning of temperature often necessitates an additional calibration set, and a low temperature also limits the reasoning capability, degrading the LLM performance.

In addition, our empirical results show this issue is further enlarged, as the increase of the reasoning difficulty. To this end, we explore a new problem, namely, *Resource-Constrained Confidence Estimation for LLM Reasoning*, where the sample size is limited and additional calibration set is unavailable. The constraints in sampling size and additional dataset is very practical in LLM reasoning tasks, especially considering huge computational overhead (Zhou et al., 2024) and the expensive labeling cost (Lightman et al., 2023). Hence, our goal is to estimate accurate self-consistency confidence using given samples to achieve better reasoning performance.

To address this problem, we propose to integrate the prediction probability of LLMs into the self-consistency confidence estimation, which forms our **Perplexity Consistency** confidence estimation approach (PC). The rationale of this approach lies in that using the accurate LLM prediction probability with zero variance instead of the crude Monte-Carlo sampling can significantly reduce the variance. In addition, theoretical analysis also confirms that the integration could indeed reduce the estimation error and achieves a quadratic $\mathcal{O}(1/n^2)$ decreasing rate. Moreover, the decomposition of estimation error guides us to further boost the convergence rate to be exponential in $n$, through pruning the low probability reasoning paths. To achieve this, we propose *Reasoning Pruning* to model the confidence distribution and automatically remove the reasoning paths with low probability. Combining PC approach and *Reasoning Pruning*, we build our **Reasoning-pruning Perplexity Consistency** confidence estimation approach (RPC). Our experimental results on seven benchmark datasets, including mathematical reasoning and code generation tasks, demonstrate that our proposed PC and RPC approaches deliver superior performance compared to existing methods.

The contributions of this paper are summarized as follows:

(1) We introduce and highlight a new problem setting, namely, *Resource-Constrained Confidence Estimation for LLM Reasoning*, which aims to accurately estimate self-consistency confidence with a constrained sample size and without additional calibration set in order to achieve improved reasoning performance.

(2) We propose the PC and RPC approaches, which leverage accurate LLM prediction probabilities to reduce the variance in crude Monte-Carlo sampling process and prune low-probability reasoning paths to achieve faster convergence, respectively.

(3) Our theoretical analysis demonstrates that PC achieves a quadratic error decreasing rate of $\mathcal{O}(1/n^2)$, which is faster than the linear rate of standard self-consistency method. Furthermore, the decomposition of the estimation error guides us in designing our approach to boost the convergence rate to be exponential in $n$.

(4) We conducted experiments on four mathematical reasoning tasks, and the results demonstrate that our PC and RPC approaches achieve significant improvements in both accuracy and calibration error. Moreover, the results from code generation tasks further confirm the generalizability of our approaches.

The remainder of this paper is organized as follows: Section 2 briefly reviews the confidence estimation methods of LLMs and some evaluation metrics for them. Section 3 reveals the self-consistency confidence estimation problem through empirical observations, followed by some discussion of the practical constraints. In Section 4, we present the estimation error reduction methods PC and RPC, and provide some theoretical analyses of their efficacy and efficiency. Section 5 reports our experimental results. We conclude the paper in Section 6.

## 2 PRELIMINARY AND RELATED WORK

### 2.1 CONFIDENCE OF TRADITIONAL MODELS

Assume a neural network model $f_{\boldsymbol{\theta}}$ for a $K$-classification task. Given any data point $(\boldsymbol{x}, y)$, we denote the logit vector predicted by the neural network as $f_{\boldsymbol{\theta}}(\boldsymbol{x})$. The *confidence* of this prediction can be computed using the Softmax function, i.e.,

$$p_{\boldsymbol{\theta}}(\hat{y} \,|\, \boldsymbol{x}) = \frac{\exp([f_{\boldsymbol{\theta}}(\boldsymbol{x})]_{\hat{y}})}{\sum_{i=1}^{K} \exp([f_{\boldsymbol{\theta}}(\boldsymbol{x})]_i)}, \tag{1}$$

where $\hat{y} = \arg\max_{i \in [K]} [f_{\boldsymbol{\theta}}(\boldsymbol{x})]_i$. The confidence of a model is considered *well-calibrated* if it reflects the true probability of its prediction being correct, i.e.,

$$\mathbb{P}_{(\boldsymbol{x},y)}\left(\hat{y} = y \,|\, p_{\boldsymbol{\theta}}(\hat{y} \,|\, \boldsymbol{x}) = c\right) = c. \tag{2}$$

To measure the calibration performance, expected calibration error (ECE) and its upper bound Brier score (BS), are usually adopted. We formulate ECE and BS as follows, where $\mathbb{I}(\cdot, \cdot)$ is an indicator function determining whether the two predictions are equal.

$$
\begin{aligned}
\text{ECE}: &\quad \mathbb{E}_c\left[\left|\mathbb{P}_{(\boldsymbol{x},y)}\left(\hat{y} = y \,|\, p_{\boldsymbol{\theta}}(\hat{y} \,|\, \boldsymbol{x}) = c\right) - c\right|\right], \\
\text{BS}: &\quad \mathbb{E}_{(\boldsymbol{x},y)}\left[\Sigma_{i=1}^K(\mathbb{I}(y_i, y) - p_{\boldsymbol{\theta}}(y_i \,|\, \boldsymbol{x}))^2\right].
\end{aligned} \tag{3}
$$

Existing studies (Guo et al., 2017) reveal that recent models are usually not well-calibrated. Thereby, several in-process methods (e.g., mix-up (Hendrycks et al., 2019), label smoothing (Müller et al., 2019), and focal loss (Mukhoti et al., 2020)) and post-hoc methods (e.g., isotonic regression (Barlow & Brunk, 1972), Platt scaling (Platt et al., 1999), and temperature scaling (Guo et al., 2017)) have been thoroughly explored to achieve a better calibration.

## 2.2 CONFIDENCE OF LLMS

Recent advances of LLMs exhibit their strong capabilities in reasoning tasks such as arithmetic (Lewkowycz et al., 2022a), commonsense (Zhao et al., 2023), and symbolic reasoning (Gao et al., 2023). Among these studies, the introduction of well-calibrated confidence can further contribute to migrate the bias and alleviate the hallucination of LLMs (Zheng et al., 2023; Bubeck et al., 2023; Geng et al., 2024). Specifically, given an LLM $p_{\boldsymbol{\theta}}(\cdot \,|\, \cdot)$ parameterized by $\boldsymbol{\theta}$, the reasoning task takes a token sequence $\boldsymbol{x}$ as input, and generates a token sequence $\hat{\boldsymbol{y}} = (t_1, t_2, \ldots, t_m)$ as the answer, where each token $t_i$ is sampled from the parametric distribution of LLM $p_{\boldsymbol{\theta}}(t_i \,|\, \boldsymbol{x}, t_{<i})$. Naïvely, we can extend the confidence from traditional models to LLMs, through simply aggregating the confidence of each output token in the answer $\hat{\boldsymbol{y}}$ (we name it token-level confidence), i.e.,

$$p_{\boldsymbol{\theta}}^{(\text{TL})}(\hat{\boldsymbol{y}} \,|\, \boldsymbol{x}) = p_{\boldsymbol{\theta}}(t_1 \,|\, \boldsymbol{x}) \cdot p_{\boldsymbol{\theta}}(t_2 \,|\, \boldsymbol{x}, t_1) \cdots p_{\boldsymbol{\theta}}(t_m \,|\, \boldsymbol{x}, t_{<m-1}). \tag{4}$$

However, such confidence is highly sensitive to the length of output token sequence, and several adaptions are proposed in literature. Next, we briefly summarize three common confidence measures for the reasoning paths generated of LLMs.

**Sentence-level Confidence (PPL).** Huang et al. (2023) and Duan et al. (2024) propose using the geometric mean version to replace the naïve token-level confidence:

$$p_{\boldsymbol{\theta}}^{(\text{SL})}(\hat{\boldsymbol{y}} \,|\, \boldsymbol{x}) = \left(p_{\boldsymbol{\theta}}(t_1 \,|\, \boldsymbol{x}) \cdot p_{\boldsymbol{\theta}}(t_2 \,|\, \boldsymbol{x}, t_1) \cdots p_{\boldsymbol{\theta}}(t_m \,|\, \boldsymbol{x}, t_{<m-1})\right)^{\frac{1}{m}}, \tag{5}$$

which is also called *perplexity* of the LLM answer (Chen et al., 1998; Blei et al., 2003).

**Self-consistency Confidence (SC).** The consistency (Wang et al., 2022; Chen et al., 2023; Cheng et al., 2024) between different generated answers, known as *self-consistency*, has been shown to be able to improve the reasoning performance of LLMs. To this end, recent work (Xiong et al., 2023; Yadkori et al., 2024; Becker & Soatto, 2024) proposes establishing LLM confidence also based on the self-consistency. To compute this confidence, an consistency function $\mathbb{I}_C(\cdot, \cdot)$ should be defined to determine the consistency between a pair of generated answers. Then, for any given input $\boldsymbol{x}$ and its associated answer $\hat{\boldsymbol{y}}$ generated by the LLM, the self-consistency method additionally builds a reference answers of size $n$, i.e., $\hat{\boldsymbol{y}}_i, \ldots, \hat{\boldsymbol{y}}_n$, sampled from LLM's parametric distribution $p_{\boldsymbol{\theta}}(\hat{\boldsymbol{y}}_i \,|\, \boldsymbol{x})$. The self-consistency confidence is computed according to the proportion of the consistency between the associated answer $\hat{\boldsymbol{y}}$ and a series of reference answers, i.e.,

$$p_{\boldsymbol{\theta}}^{(\text{SC})}(\hat{\boldsymbol{y}} \,|\, \boldsymbol{x}) = \left(\mathbb{I}_C(\hat{\boldsymbol{y}}, \hat{\boldsymbol{y}}_1) + \cdots + (\mathbb{I}_C(\hat{\boldsymbol{y}}, \hat{\boldsymbol{y}}_n)\right)/n, \tag{6}$$

There are quite a few strategies to implement the consistency function $\mathbb{I}_C(\cdot, \cdot)$ toward different tasks, such as Jaccard similarity and logical entailment in commonsense reasoning (Lin et al., 2023; Kuhn et al., 2023), numerical comparison in math problem solving (Yu et al., 2024), and execution matching (Chen et al., 2022) in code generation.

**Verbalized Confidence (VERB).** Another approach to defining LLM confidence is *verbalization* (Kadavath et al., 2022; Xiong et al., 2023; Tian et al., 2023), which directly prompts the LLM

Table 1: Performance comparison of confidence estimation methods on MATH dataset. The SC method gives the best performance on both accuracy and confidence calibration metrics.

|  | Accuracy ($\uparrow$) | ECE ($\downarrow$) | Brier Score ($\downarrow$) |
|---|---|---|---|
| PPL | 46.68 | 49.52 | 49.13 |
| VERB | 25.81 | 48.12 | 43.51 |
| Naïve SC | 42.77 | 41.47 | 33.60 |
| SC | **50.50** | **6.60** | **15.49** |

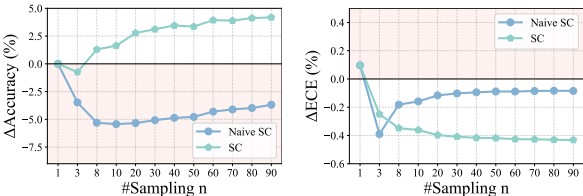

Figure 1: The accuracy and ECE gaps of SC and naïve SC methods on MATH dataset. The red area in figure indicates performance below the baseline.

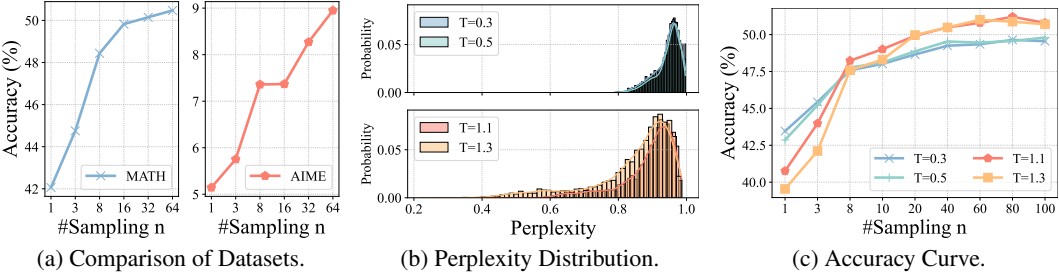

(a) Comparison of Datasets.  (b) Perplexity Distribution.  (c) Accuracy Curve.

Figure 2: Several factors impact the variance of SC method. (a) shows that performance converges more slowly with $n$ on difficult dataset; (b) shows that high temperatures offer large sampling space measured by perplexity; (c) shows that high temperatures provide better performance upper bound.

to express the confidence level $p_{\boldsymbol{\theta}}^{(\text{VB})}(\hat{\boldsymbol{y}} \mid \boldsymbol{x})$ alongside the output answer $\hat{\boldsymbol{y}}$ (e.g., "Read the question, provide your answer, and your confidence in this answer."). Existing work demonstrates that the verbalized confidence can be further improved with proper instruction fine-tuning (Mielke et al., 2022; Lin et al., 2022; Zhang et al., 2024). In addition, there are a series of variants to the verbalization method, including multi-agent deliberation (Yang et al., 2024), top-k ranking (Tian et al., 2023), few-shot prompting (Liu et al., 2023), and reflection (Dhuliawala et al., 2023; Zhao et al., 2024). Particularly, to integrate verbalization with chain-of-thought prompting in reasoning tasks, a multi-step version has been developed (Xiong et al., 2023). This version initially assigns confidence levels to individual reasoning steps and then aggregates these to form the overall confidence.

## 3 PROBLEM ANALYSIS

**Empirical observations.** We first evaluate the performance of the existing calibration metrics for LLM reasoning. Specifically, we use the MATH dataset (Hendrycks et al., 2021b) with InternLM-2-MATH-Plus 7B model and standard temperature (T = 1.0). Note that BS involves the enumeration of all outputs, which is virtually impractical for LLMs reasoning, and thus we replace enumeration by sampling and summation by expectation:

$$\text{BS} : \mathbb{E}_{(\boldsymbol{x}, \boldsymbol{y})} \mathbb{E}_{\tilde{\boldsymbol{y}}} \left[ (\mathbb{I}_S(\tilde{\boldsymbol{y}}, \boldsymbol{y}) - p_{\boldsymbol{\theta}}(\tilde{\boldsymbol{y}} \mid \boldsymbol{x}))^2 \right],$$

where $\hat{\boldsymbol{y}}$ is also sampled from LLM parametric distribution $p_{\boldsymbol{\theta}}(\hat{\boldsymbol{y}} \mid \boldsymbol{x})$; $\mathbb{I}_S(\cdot, \cdot)$ denotes the semantic equivalence between two answers, i.e., whether the two answers have the same meanings although they may be expressed in different forms. For the consistency function $\mathbb{I}_C(\cdot, \cdot)$ in SC, we instantiate it using the answer comparison (Yu et al., 2024), which is commonly used in math reasoning. As an ablative version, we also define a naïve version of SC by using string comparison instead, denoted as Naïve SC method.

Results shown in Table 1 demonstrate that the self-consistency confidence SC achieves the best performance across accuracy and calibration metrics. This observation is also confirmed by our detailed results in Appendix C.2. In addition, we use the sentence-level confidence PPL as a baseline, and compute the accuracy gap and the ECE gap from the baseline to Naïve SC and SC with various sample sizes $n$. The results shown in Figure 1 illustrate that SC, using an appropriate consistency function, can consistently outperform PPL when $n$ is sufficient.

**Challenges.** Although promising, SC's performance faces a dilemma. On one hand, its performance could be significantly affected by the sampling sufficiency of reference answer. For example, as shown in Figure 2a, the requirements of sample size dramatically grows as the reasoning problems become much harder (e.g., MATH→AIME). On the other hand, constructing a large collection of reference answers is often computationally expensive and time-consuming. For example, generating a 100-sized reference answers for all problems in MATH, which contains 5,000 problems, using a single A800 GPU would take about 18 hours. The total emissions are estimated to be 4.14 kg $CO_2$ when using Google Cloud Platform in the asia-south1 region (Lacoste et al., 2019). This dilemma raises a key question: *Is it possible to accurately estimate self-consistency confidence, even when the sample size is limited?*

Since the sampling estimation in self-consistency confidence is unbiased, the error from insufficient sampling is mainly caused by variance. The above question then becomes: *How to effectively control the variance of self-consistency confidence, particularly when the sample size is limited?*

To answer the above question, a straightforward method to control the variance could be choosing a conservative temperature, which tends to narrow the sampling space of LLMs. To illustrate this, we present the perplexity distribution at different temperatures in Figure 2b. However, as evident in Figure 2c, although a low sampling temperature successfully decreases the induced variance, it sacrifices model performance. In other words, a high temperature T offers better performance potential but requires a larger sample size $n$, creating another dilemma between accuracy and efficiency. Moreover, selecting an appropriate temperature T requires an additional calibration set and computational resources (Guo et al., 2017), which is expensive in practice.

To this end, our analysis motivates us to study a novel and challenging problem setting for LLM reasoning, namely, *Resource-Constrained Confidence Estimation for LLM Reasoning*. In this setting, we aim to achieve better confidence estimation even when the sampled reference collection is of small size, thereby improving both accuracy and reducing calibration error in LLM reasoning tasks.

## 4 METHODOLOGY

### 4.1 VARIANCE REDUCTION

For given input $x$ and its associated prediction $\hat{y}$, we define the oracle self-consistency confidence by $\psi$. Formally, the oracle $\psi$ is the cumulative probabilities of all generated answers that are consistent to the prediction $\hat{y}$. Since a closed-form expression of the oracle $\psi$ is computationally infeasible, existing self-consistency confidence $p_{\theta}^{(SC)}(\hat{y}\,|\,x) = (\mathbb{I}_C(\hat{y}, \hat{y}_1) + \cdots + (\mathbb{I}_C(\hat{y}, \hat{y}_n))/n$ is essentially a Monte-Carlo sampling estimation to the oracle $\psi$. Hence, the expectation is unbiased, and the variance can be computed by $\frac{\psi(1-\psi)}{n}$, which exhibits a linear convergence rate of the sample size $n$.

We propose to boost the variance reduction efficiency by directly using the prediction probability of LLMs, rather than the crude sampling estimation. To achieve this, we first define $\mathbb{I}_T(\cdot, \cdot)$ as the token-level consistency (i.e., string comparison). Then, by viewing $\mathbb{I}_T$ as a self-consistency function, we can bridge the token-level confidence and its Monte-Carlo sampling estimation by

$$p_{\theta}^{(TL)}(\hat{y}\,|\,x) \approx (\mathbb{I}_T(\hat{y}, \hat{y}_1) + \cdots + (\mathbb{I}_T(\hat{y}, \hat{y}_n))/n.$$

Now, the self-consistency confidence estimation can be reformulated as

$$p_{\theta}^{(SC)}(\hat{y}|x) = \frac{1}{n}\left(\mathbb{I}_C(\hat{y}, \hat{y}_1) + \cdots + \mathbb{I}_C(\hat{y}, \hat{y}_n)\right)$$

$$= \sum_{\tilde{y} \in \text{set}(\hat{y}_1, \ldots, \hat{y}_n)} \left[\mathbb{I}_C(\hat{y}, \tilde{y}) \cdot \frac{\sum_{i=1}^n \mathbb{I}_T(\tilde{y}, y_i)}{n}\right]$$

$$\approx \sum_{\tilde{y} \in \text{set}(\hat{y}_1, \ldots, \hat{y}_n)} \left[\mathbb{I}_C(\hat{y}, \tilde{y}) \cdot p_{\theta}^{(TL)}(\tilde{y}|x)\right],$$

where $\text{set}(\hat{y}_1, \ldots, \hat{y}_n)$ denotes the set of reference answers, which remove the duplicate ones from the original reference collection. We call this new version of confidence estimation by *Perplexity Consistency Confidence* (PC), as it tends to combine token-level and self-consistency confidences.

For ease of notation, we denote the original self-consistency by $\hat{\psi}_1$; and denote the perplexity consistency confidence by $\hat{\psi}_2$, i.e.,

$$\hat{\psi}_1 = \sum_{i=1}^{n} \frac{\mathbb{I}_C(\hat{\boldsymbol{y}}, \hat{\boldsymbol{y}}_i)}{n}, \quad \hat{\psi}_2 = \sum_{\tilde{\boldsymbol{y}} \in \mathrm{set}(\hat{\boldsymbol{y}}_1, \ldots, \hat{\boldsymbol{y}}_n)} \mathbb{I}_C(\hat{\boldsymbol{y}}, \tilde{\boldsymbol{y}}) \cdot p_{\boldsymbol{\theta}}^{(\mathrm{TL})}(\tilde{\boldsymbol{y}}|\boldsymbol{x}).$$

We have the following two theorems, which compare the estimation errors between $\hat{\psi}_1$ and $\hat{\psi}_2$.

**Theorem 1** (Pc achieves lower estimation error than Sc). *Given a fixed LLM with parameters $\boldsymbol{\theta}$, an input $\boldsymbol{x}$ and any output $\hat{\boldsymbol{y}}$. With a proper assumption,* Pc *confidence estimation has a lower error than* Sc *confidence estimation:*

$$\mathbb{E}[(\hat{\psi}_1 - \psi)^2] \leq \mathbb{E}[(\hat{\psi}_2 - \psi)^2].$$

*Remarks.* The proof is detailed in Appendix A.1. The theorem states that Pc possess a lower estimation error when $n$ is limited. We also carry out a synthesis experiment of this theorem in Appendix A.3, which further illustrates the validity of required condition and the derived effectiveness.

**Theorem 2** (Pc achieves higher convergence rate than Sc). *Given a fixed LLM with parameters $\boldsymbol{\theta}$, an pair of input $\boldsymbol{x}$ and output $\hat{\boldsymbol{y}}$, and a $n$-sized reference answers $\hat{\boldsymbol{y}}_1, \ldots, \hat{\boldsymbol{y}}_n$. Then,* Pc *confidence estimation error decreases with a quadratic rate $\mathcal{O}(1/n^2)$; while* Sc *confidence estimation error decreases with a linear rate $\mathcal{O}(1/n)$.*

*Remarks.* The proof is presented in Appendix A.2. The theorem illustrates that Pc provides a higher-order convergence rate in the sense of estimation error, indicating the performance improvement can be consistently enlarged with $n$ increased.

## 4.2 Reasoning Pruning

The following theorem gives an in-depth analysis of convergence rate in estimation error, revealing a clue for further improving the variance reduction effectiveness.

**Theorem 3** (The decomposition of Pc estimation error). *Given a fixed LLM with parameters $\boldsymbol{\theta}$, an input $\boldsymbol{x}$ and any output $\hat{\boldsymbol{y}}$. Let the answer space be divided into two parts $\Omega_1 = \{\tilde{\boldsymbol{y}} \mid p_{\boldsymbol{\theta}}^{(TL)}(\tilde{\boldsymbol{y}}|\boldsymbol{x}) \leq n^{(-\frac{r}{2})}\}$ and $\Omega_2 = \{\tilde{\boldsymbol{y}} \mid p_{\boldsymbol{\theta}}(\tilde{\boldsymbol{y}}|\boldsymbol{x}) > n^{(-\frac{r}{2})}\}$. Then, the upper bound to the confidence estimation error of* Sc *can be decomposed into two parts, i.e.,*

$$\mathbb{E}[(\hat{\psi}_2 - \psi)^2] \leq \sum_{\hat{\boldsymbol{y}} \in \Omega_1} \mathcal{O}(n^{-r}) + \sum_{\hat{\boldsymbol{y}} \in \Omega_2} \mathcal{O}(e^{(1-\frac{r}{2})}).$$

*Remarks.* The proof is shown in Appendix A.2. The theorem indicates that the convergence rate of variance reduction mainly hindered by generated answers in low token-level confidence region. Upon these answers are removed, the convergence rate becomes exponential in $n$, which significantly superior than the quadratic rate.

This theorem motivates us to propose the *Reasoning Pruning* to refine the reference collections by removing reasoning paths in low token-level confidence region. However, determining the appropriate threshold for reasoning path removal is challenging in practice, making an automated removal strategy highly desirable. Inspired by studies on open-set recognition (Bendale & Boult, 2016), we assume that the token-level confidence distribution of each reference can be modeled as a mixture of two Weibull distributions, representing high and low token-level confidence, respectively. The probability density function (PDF) for the mixture is:

$$f(x) = w_1 \cdot f_{\mathrm{Weibull}}(x; k_1, \lambda_1) + w_2 \cdot f_{\mathrm{Weibull}}(x; k_2, \lambda_2). \tag{7}$$

where Weibull PDF is $f_{\mathrm{Weibull}}(x; k, \lambda) = \frac{k}{\lambda} \left(\frac{x}{\lambda}\right)^{k-1} \exp\left(-\left(\frac{x}{\lambda}\right)^k\right)$. Maximum likelihood estimation (MLE) is employed to estimate the parameters, i.e., $(k_1, \lambda_1)$, $(k_2, \lambda_2)$, $w_1$, and $w_2$. We denote Weibull$(k_1, \lambda_1)$ as the high confidence distribution and Weibull$(k_2, \lambda_2)$ as the low confidence distribution.

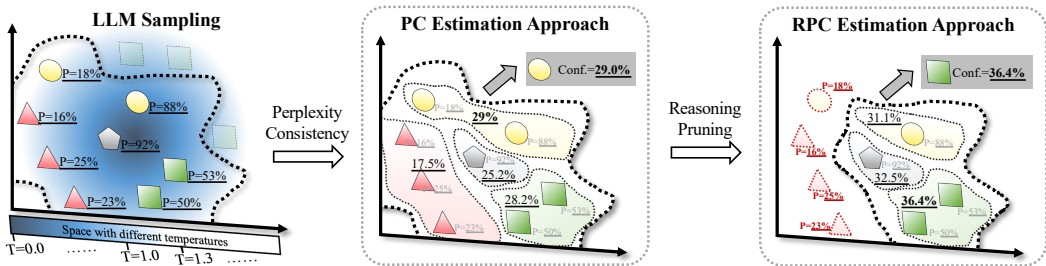

Figure 3: The illustration of the PC and RPC approaches. PC integrates the LLM probability into self-consistency confidence estimation to reduce variance. RPC additionally remove the low-probability answers to achieve a faster convergence rate of estimation error.

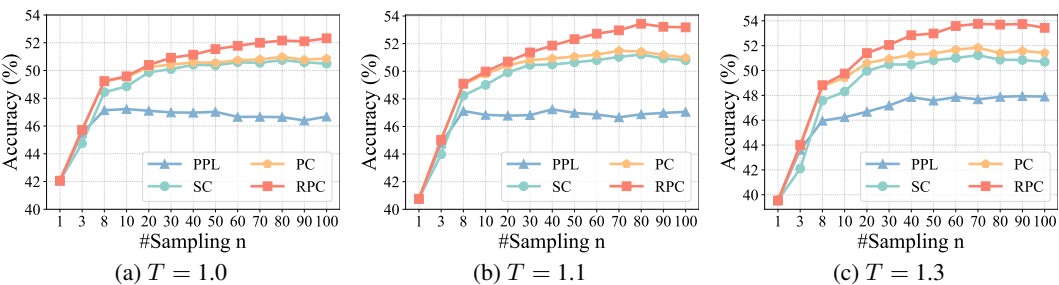

| (a) $T = 1.0$ | (b) $T = 1.1$ | (c) $T = 1.3$ |

Figure 4: The accuracy of the InternLM-2-MATH-Plus 7B model on MATH dataset, with different sample size $n$ and temperature T. Our proposed PC and RPC approaches consistently achieved the best performance in all scenarios.

Then, for each output $\hat{y}$ with confidence $\hat{c} = p_{\theta}^{(\text{TL})}(\hat{y}; x)$, we can compute its probability belonging to the high confidence distribution Weibull($k_1, \lambda_1$) using Bayes' theorem:

$$P_{Rel}(\hat{c}) = \frac{w_1 \cdot f_{\text{Weibull}}(\hat{c}; k_1, \lambda_1)}{w_1 \cdot f_{\text{Weibull}}(\hat{c}; k_1, \lambda_1) + w_2 \cdot f_{\text{Weibull}}(\hat{c}; k_2, \lambda_2)}. \tag{8}$$

We remove reasoning paths with token-level confidence $\hat{c}$ that satisfies $P_{Rel}(\hat{c}) > 1 - P_{Rel}(\hat{c})$. Moreover, to ensure the stability of the algorithm when $n$ is limited, we employ the Truncated Mean method (Marazzi & Ruffieux, 1999), retaining outputs with token-level confidence greater than the overall mean. This prevents the removal of too many reasoning paths due to potential inaccurate estimation of the mixture distribution.

We apply the *Reasoning Pruning* to the reference collection and then compute the confidence based on *Perplexity Consistency*, forming our proposed **R**easoning-pruning **P**erplexity **C**onsistency confidence RPC. The overall illustration of PC and RPC confidences are presented in the Figure 3.

## 5 EXPERIMENTS

### 5.1 EXPERIMENTAL SETTING

**Comparison Methods.** We compare three types of LLM confidences: sentence-level confidence (Perplexity, PPL (Wang et al., 2022)), self-consistency confidence (SC (Chen et al., 1998)), and verbalized confidence (VERB (Tian et al., 2023)). For mathematical reasoning tasks, the verbalized confidence is computed based on the probability that the LLM responds "True" versus "False" when asked an "Is-True" question. For code generation tasks, we extracted verbalized confidence scores from model's numerical likelihood expressions by prompting the LLM.

**Datasets.** For mathematical reasoning tasks, we evaluate each method on one common mathematical benchmark datasets, i.e., MATH (Hendrycks et al., 2021b), and three challenging mathematical

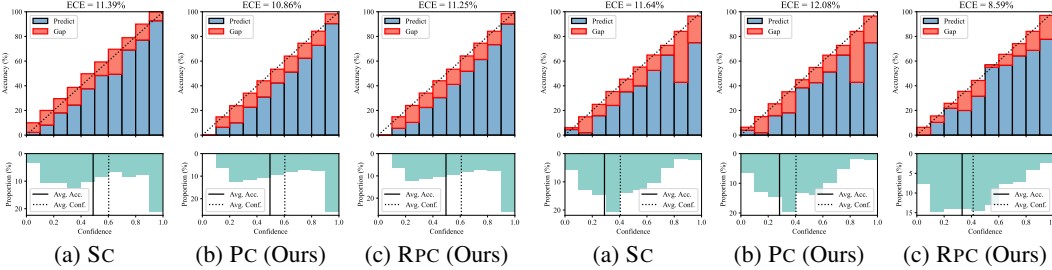

| (a) Sc | (b) Pc (Ours) | (c) Rpc (Ours) | (a) Sc | (b) Pc (Ours) | (c) Rpc (Ours) |

Figure 5: The ECE of InternLM-2-MATH-Plus 7B model on MATH dataset with $T = 1.0$ and $n = 10$. Both Pc and Rpc give satisfied ECE and Rpc gives the minimal ECE.

Figure 6: The ECE of InternLM-2-MATH-Plus 7B model on MathOdyssey dataset with $T = 1.0$ and $n = 256$. Rpc gives the significant better ECE than Pc method.

Table 2: The accuracy and ECE of the InternLM-2-MATH-Plus 7B model on MATH and three Olympiad-level datasets using non-conservative temperatures ($T \in \{1.0, 1.1, 1.3\}$). The best performance is highlighted in **bold**, while the second-best performance is underlined.

| Methods | T = 1.0 | | | T = 1.1 | | | T = 1.3 | | |
|---|---|---|---|---|---|---|---|---|---|
| | Odyssey | OlympiadBench | AIME | Odyssey | OlympiadBench | AIME | Odyssey | OlympiadBench | AIME |
| Ppl | 25.45 | 7.17 | 5.14 | 27.25 | 6.93 | 5.89 | 25.45 | 7.55 | 6.65 |
| Verb | 9.37 | 3.49 | 3.13 | 9.32 | 3.88 | 3.17 | 8.39 | 3.23 | 2.27 |
| Sc | 28.92 | 11.06 | 9.54 | 28.41 | 10.79 | 8.57 | 27.63 | 10.40 | 8.20 |
| Pc | 28.28 | 11.06 | 9.65 | 28.79 | 10.83 | 8.68 | 28.02 | 10.67 | 8.68 |
| Rpc | **33.16** | **11.21** | **9.75** | **34.19** | **11.14** | **9.75** | **32.13** | **11.29** | **8.90** |

datasets that include Olympiad-level problems, i.e., MathOdyssey (Fang et al., 2024), Olympiad-Bench (He et al., 2024), and AIME (Zamil & Rabby, 2024). For code generation tasks, we evaluate each method on three benchmark datasets, i.e., HumanEval (Chen et al., 2021), MBPP (Austin et al., 2021), and introductory-level problems of APPS (Hendrycks et al., 2021a). We presents the details of each dataset in Appendix B.1.

**Implementation Details.** For mathematical reasoning tasks, we evaluate the InternLM2-Math-Plus models with 1.8B and 7B parameters (Ying et al., 2024), as well as the DeepSeekMath-RL 7B model (Shao et al., 2024). The consistency function $\mathbb{I}_C$ is answer comparison. For code generation tasks, we evaluate the Deepseek-Coder 33B model. The consistency function $\mathbb{I}_C$ is constructed based on semantic equivalence (Malík & Vojnar, 2021) by clustering code base on given test cases. All experiments were conducted on Linux servers equipped with A800 and H800 GPUs.

## 5.2 Experimental Results

**Results on MATH Dataset.** We first conduct experiments on the MATH dataset using InternLM-2-MATH-Plus 7B model with non-conservative temperatures ($T \in \{1.0, 1.1, 1.3\}$) and various sample sizes $n$. Pc and Rpc approaches give better accuracy compared to Sc and Ppl methods in Figure 4. The ECE in Figure 5 also demonstrates the superior performance of Rpc approach with a limited sample size of $n = 10$. These results demonstrate that our *Perplexity Consistency* method effectively reduces the variance of Sc, leading to more reliable confidence estimation. Moreover, our Rpc approach shows a significant performance improvement compared to Pc, as shown in Figure 4. The performance gap between Rpc and Pc demonstrates that *Reasoning Pruning* effectively mitigates the variance introduced by non-conservative sampling temperatures, allowing Rpc approach to estimate reliable confidence using non-conservative sampling temperatures in practice. Overall, the Rpc approach is proved to be practical for real-world reasoning tasks, and we recommend employing non-conservative sampling temperatures when utilizing the Rpc approach.

**Results on Difficult Mathematics Datasets.** To further access the effectiveness of Pc and Rpc, we evaluate each method on three challenging datasets that include Olympiad-level problems. For each problem, we sample 256 solutions using three non-conservative temperatures, i.e., $T \in \{1.0, 1.1, 1.3\}$. Table 2 shows that our Rpc consistently achieves the highest accuracy across

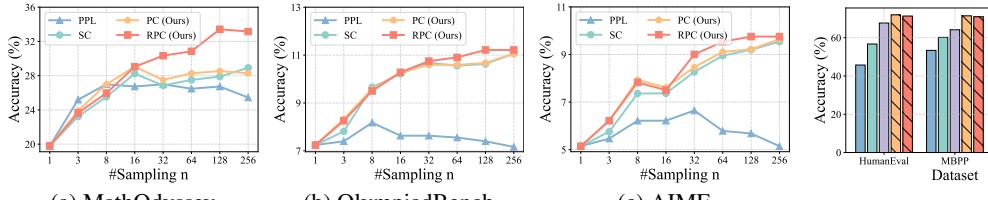

(a) MathOdyssey  (b) OlympiadBench  (c) AIME

Figure 7: The accuracy of InternLM-2-MATH-Plus 7B model on hard datasets across different number of samplings $n$ using sampling temperature $T = 1.0$. RPC gives the best performance in major cases.

Figure 8: Accuracy of DeepSeek-Coder 33B model on code generation benchmarks.

Table 3: The accuracy of two different models on the four math reasoning datasets. The best performance is highlighted in **bold**, while the second-best performance is underlined.

| Methods | InternLM2-Math-Plus 1.8B | | | | DeepSeek-Math 7B | | | |
|---|---|---|---|---|---|---|---|---|
| | MATH | MathOdyssey | OlympiadBench | AIME | MATH | MathOdyssey | OlympiadBench | AIME |
| PPL | 32.44 | 15.94 | 2.26 | 1.39 | 41.52 | 21.59 | 5.22 | 2.89 |
| VERB | 6.40 | 2.27 | 0.58 | 0.20 | 13.02 | 1.76 | 2.09 | 1.70 |
| SC | 36.61 | 14.40 | 6.07 | 2.68 | 53.54 | 36.25 | 11.49 | 9.36 |
| PC | 36.88 | 14.65 | 6.07 | 2.68 | 53.56 | 36.25 | 11.45 | 9.65 |
| RPC | **38.16** | **15.68** | **6.54** | **3.43** | **53.58** | **37.28** | **11.60** | **9.86** |

all datasets and sampling temperatures, while PC consistently ranks second. The accuracy results across various sampling sizes in Figure 7 and the ECE results in Figure 6 further demonstrate the superiority of the proposed approaches. These results demonstrate that the proposed approaches, especially the RPC approach, perform well on challenging mathematical reasoning datasets, enhancing model performance by providing more accurate confidence.

**Results of Different Models.** To evaluate whether our approaches can generalize to different scales and types of models, we conduct experiments on InternLM2-Math-Plus 1.8B and DeepSeek-Math 7B models following the same setting using $T = 1.0$. The results in Table 3 demonstrate that proposed approaches, especially RPC, consistently outperform existing methods.

**Results on Code Generation Tasks.** To investigate whether our proposed approaches can generalize to other tasks, e.g., code generation tasks, we evaluate our approaches and comparison methods on three code generation benchmarks, as shown in Figure 8. The results show that the PC approach achieves the best accuracy across all datasets. The *Reasoning Pruning* in RPC did not improve performance, because low-probability code is often incurs compilation errors, which are previously removed in the code evaluation process. Despite this, RPC still ranks second with minimal performance loss, demonstrating its robustness.

## 6 CONCLUSION

In this paper, we explore *Resource-Constrained Confidence Estimation for LLM Reasoning*, where the goal is to estimate accurate self-consistency confidence with a limited sample size to achieve better reasoning performance. We integrate the probability directly obtained from LLMs to reduce the variance of crude Monte-Carlo sampling estimation method, forming the PC approach. Our theoretical analysis guarantees that the estimation error decreases at a quadratic rate of $\mathcal{O}(1/n^2)$. Additionally, we introduce a *Reasoning Pruning* strategy, motivated by our decomposed estimation error, to further boost the convergence rate to exponential in $n$. Experiments on four mathematical reasoning tasks demonstrate the effectiveness of the proposed PC and RPC approaches. Furthermore, results from code generation tasks highlight the generalizability of our methods.

One limitation of this work is that we were unable to explore additional mathematical models with varying parameter scales due to resource constraints. However, we believe that our current experiments are robust enough to validate our claims and demonstrate the superior performance of our proposed approaches.

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

## A  THEORETICAL RESULTS

Given an LLM parameterized by $\boldsymbol{\theta}$, we first sample $n$ reference answers for the input $\boldsymbol{x}$, denoted by $\hat{\boldsymbol{y}}_i, i = 1, \dots, n$. For the LLM prediction output $\hat{\boldsymbol{y}}$, we define its token-level confidence as $\phi = p_{\boldsymbol{\theta}}^{(\mathrm{TL})}(\hat{\boldsymbol{y}}|\boldsymbol{x})$, and the ground-truth self-consistency confidence as $\psi$. We have two estimators for $\psi$, where the sampling estimation and the variance-reduction version are defined by

$$\hat{\psi}_1 = \sum_{i=1}^{n} \frac{\mathbb{I}_C(\hat{\boldsymbol{y}}, \hat{\boldsymbol{y}}_i)}{n}, \quad \hat{\psi}_2 = \sum_{\tilde{\boldsymbol{y}} \in \mathrm{set}(\hat{\boldsymbol{y}}_1, \dots, \hat{\boldsymbol{y}}_n)} \mathbb{I}_C(\hat{\boldsymbol{y}}, \tilde{\boldsymbol{y}}) \cdot p_{\boldsymbol{\theta}}^{(\mathrm{TL})}(\tilde{\boldsymbol{y}}|\boldsymbol{x}).$$

Now, we start to compute the variances of these two estimators. Since reference answers are i.i.d, and $\mathbb{I}_C(\hat{\boldsymbol{y}}, \hat{\boldsymbol{y}}_i) \sim \mathrm{Bernoulli}(\psi)$, the expectation and variance of $\hat{\psi}_1$ are

$$\mathbb{E}[\hat{\psi}_1] = \psi, \quad \mathrm{Var}[\hat{\psi}_1] = \frac{\psi(1 - \psi)}{n}.$$

As to the variance-reduction version, we first rewrite it as

$$\hat{\psi}_2 = \sum_{\tilde{\boldsymbol{y}} \in \mathrm{set}(\hat{\boldsymbol{y}}_1, \dots, \hat{\boldsymbol{y}}_n)} \mathbb{I}_C(\hat{\boldsymbol{y}}, \tilde{\boldsymbol{y}}) \cdot p_{\boldsymbol{\theta}}^{(\mathrm{TL})}(\tilde{\boldsymbol{y}}|\boldsymbol{x})$$

$$= \sum_{\tilde{\boldsymbol{y}} \in \Omega_C} \mathbb{I}(\tilde{\boldsymbol{y}} \in \mathrm{set}(\hat{\boldsymbol{y}}_1, \dots, \hat{\boldsymbol{y}}_n)) \cdot p_{\boldsymbol{\theta}}^{(\mathrm{TL})}(\tilde{\boldsymbol{y}}|\boldsymbol{x}),$$

where $\Omega_C$ denotes the set of the reference answers that are consistent to $\hat{\boldsymbol{y}}$. We denotes the token-level confidence of $\hat{\boldsymbol{y}}$ by $\phi(\hat{\boldsymbol{y}})$. Based on these definitions, we have the following property holds.

$$\psi = \sum_{\tilde{\boldsymbol{y}} \in \Omega_C} \phi(\tilde{\boldsymbol{y}})$$

Then, its expectation can be computed by

$$\mathbb{E}[\hat{\psi}_2] = \sum_{\tilde{\boldsymbol{y}} \in \Omega_C} (1 - (1 - \phi(\tilde{\boldsymbol{y}}))^n) \cdot \phi(\tilde{\boldsymbol{y}})$$

Next, we can compute the expectation of squared confidence by

$$\mathbb{E}[\hat{\psi}_2^2] = \sum_{\tilde{\boldsymbol{y}} \in \Omega_C} (1 - (1 - \phi(\tilde{\boldsymbol{y}}))^n) \cdot \phi(\tilde{\boldsymbol{y}})^2$$

$$+ \sum_{\tilde{\boldsymbol{y}}_1, \tilde{\boldsymbol{y}}_2 \in \Omega_C, \tilde{\boldsymbol{y}}_1 \neq \tilde{\boldsymbol{y}}_2} (1 - (1 - \phi(\tilde{\boldsymbol{y}}_1))^n)(1 - (1 - \phi(\tilde{\boldsymbol{y}}_2))^n) \cdot \phi(\tilde{\boldsymbol{y}}_1)\phi(\tilde{\boldsymbol{y}}_2)$$

Putting together, we derive the variance as

$$\mathrm{Var}[\hat{\psi}_2] = \sum_{\tilde{\boldsymbol{y}} \in \Omega_C} (1 - (1 - \phi(\tilde{\boldsymbol{y}}))^n) \cdot \phi(\tilde{\boldsymbol{y}})^2 - \sum_{\tilde{\boldsymbol{y}} \in \Omega_C} (1 - (1 - \phi(\tilde{\boldsymbol{y}}))^n)^2 \cdot \phi(\tilde{\boldsymbol{y}})^2$$

$$= \sum_{\tilde{\boldsymbol{y}} \in \Omega_C} (1 - (1 - \phi(\tilde{\boldsymbol{y}}))^n)(1 - \phi(\tilde{\boldsymbol{y}}))^n \cdot \phi(\tilde{\boldsymbol{y}})^2.$$

Using the bias-variance decomposition, we can compute the estimation error by

$$\mathbb{E}[(\hat{\psi}_2 - \psi)^2] = (\mathbb{E}[\hat{\psi}_2] - \psi)^2 + \mathrm{Var}[\hat{\psi}_2]$$

$$= (\sum_{\tilde{\boldsymbol{y}} \in \Omega_C} (1 - \phi(\tilde{\boldsymbol{y}}))^n \cdot \phi(\tilde{\boldsymbol{y}}))^2 + \sum_{\tilde{\boldsymbol{y}} \in \Omega_C} (1 - (1 - \phi(\tilde{\boldsymbol{y}}))^n)(1 - \phi(\tilde{\boldsymbol{y}}))^n \cdot \phi(\tilde{\boldsymbol{y}})^2$$

### A.1  PROOF OF THEOREM 1

*Proof.* We define the $\alpha$ as the minimal probability of all generated answers that are consistent to the prediction $\hat{\boldsymbol{y}}$. Next, we should that $\hat{\psi}_2$ is a tighter estimation of $\psi$ compared to $\hat{\psi}_1$ with the

assumption $n(\alpha^{2n} + \alpha^n) \leq \frac{1-\psi}{\psi}$ holds. The estimation error of $\hat{\psi}_2$ can be computed by

$$\mathbb{E}[(\hat{\psi}_2 - \psi)^2] \leq ( \sum_{\tilde{\boldsymbol{y}} \in \Omega_C} (1 - \phi(\tilde{\boldsymbol{y}}))^n \cdot \phi(\tilde{\boldsymbol{y}}))^2 + \sum_{\tilde{\boldsymbol{y}} \in \Omega_C} (1 - \phi(\tilde{\boldsymbol{y}}))^n \cdot \phi(\tilde{\boldsymbol{y}})^2$$

$$= \sum_{\tilde{\boldsymbol{y}}_1 \in \Omega_C} \sum_{\tilde{\boldsymbol{y}}_2 \in \Omega_C} (1 - \phi(\tilde{\boldsymbol{y}}_1))^n (1 - \phi(\tilde{\boldsymbol{y}}_2))^n \cdot \phi(\tilde{\boldsymbol{y}}_1)\phi(\tilde{\boldsymbol{y}}_2) + \sum_{\tilde{\boldsymbol{y}} \in \Omega_C} (1 - \phi(\tilde{\boldsymbol{y}}))^n \cdot \phi(\tilde{\boldsymbol{y}})^2$$

$$\leq \alpha^{2n} \sum_{\tilde{\boldsymbol{y}}_1 \in \Omega_C} \phi(\tilde{\boldsymbol{y}}_1) \sum_{\tilde{\boldsymbol{y}}_2 \in \Omega_C} \phi(\tilde{\boldsymbol{y}}_2) + \alpha^n \psi \sum_{\tilde{\boldsymbol{y}} \in \Omega_C} \phi(\tilde{\boldsymbol{y}})$$

$$= (\alpha^{2n} + \alpha^n)\psi^2$$

According to the assumption, we have $n(\alpha^{2n} + \alpha^n) \leq \frac{1-\psi}{\psi}$. Rearrange the inequality, we obtain that $\psi^2(\alpha^{2n} + \alpha^n) \leq \frac{(1-\psi)\psi}{n}$. Therefore, the comparison $\mathbb{E}[(\hat{\psi}_2 - \psi)^2] \leq \mathbb{E}[(\hat{\psi}_1 - \psi)^2]$ holds. $\square$

## A.2 PROOF OF THEOREM 2 AND THEOREM 3

*Proof.* To analyze the upper bound to the variance, we decompose the set $\Omega_C$ into two parts, i.e.,

$$\Omega_C = \Omega_1 \cup \Omega_2 = \{\tilde{\boldsymbol{y}} \in \Omega_C \mid \phi(\tilde{\boldsymbol{y}}) \leq n^{(-\frac{r}{2})}\} \cup \{\tilde{\boldsymbol{y}} \in \Omega_C \mid \phi(\tilde{\boldsymbol{y}}) > n^{(-\frac{r}{2})}\}.$$

Hence, we can obtain that the squared bias can be computed by

$$(\mathbb{E}[\hat{\psi}_2] - \psi)^2 = ( \sum_{\tilde{\boldsymbol{y}} \in \Omega_1} (1 - \phi(\tilde{\boldsymbol{y}}))^n \cdot \phi(\tilde{\boldsymbol{y}}) + \sum_{\tilde{\boldsymbol{y}} \in \Omega_2} (1 - \phi(\tilde{\boldsymbol{y}}))^n \cdot \phi(\tilde{\boldsymbol{y}}))^2$$

$$< ( \sum_{\tilde{\boldsymbol{y}} \in \Omega_1} n^{(-\frac{r}{2})} + \sum_{\tilde{\boldsymbol{y}} \in \Omega_2} e^{n^{(1-\frac{r}{2})}})^2.$$

The variance can be computed by

$$\mathrm{Var}[\hat{\psi}_2] \leq \frac{1}{4} \sum_{\tilde{\boldsymbol{y}} \in \Omega_1} n^{-r} + \sum_{\tilde{\boldsymbol{y}} \in \Omega_2} \psi \frac{1}{e^{(1-\frac{r}{2})}}$$

where for the first part, we have

$$\sum_{\tilde{\boldsymbol{y}} \in \Omega_1} (1 - (1 - \phi(\tilde{\boldsymbol{y}}))^n)(1 - \phi(\tilde{\boldsymbol{y}}))^n \cdot \phi(\tilde{\boldsymbol{y}})^2 \leq \frac{1}{4} \sum_{\tilde{\boldsymbol{y}} \in \Omega_1} \phi(\tilde{\boldsymbol{y}})^2 \leq \frac{1}{4} \sum_{\tilde{\boldsymbol{y}} \in \Omega_1} n^{-r}$$

and for the second part, we have

$$\sum_{\tilde{\boldsymbol{y}} \in \Omega_2} (1 - (1 - \phi(\tilde{\boldsymbol{y}}))^n)(1 - \phi(\tilde{\boldsymbol{y}}))^n \cdot \phi(\tilde{\boldsymbol{y}})^2 \leq \sum_{\tilde{\boldsymbol{y}} \in \Omega_2} \phi(\tilde{\boldsymbol{y}})^2 \frac{1}{\exp(n\phi(\tilde{\boldsymbol{y}}))} < \sum_{\tilde{\boldsymbol{y}} \in \Omega_2} \psi \frac{1}{e^{(1-\frac{r}{2})}}$$

Putting together, we can derive that the estimation error is bounded by

$$\mathbb{E}[(\hat{\psi}_2 - \psi)^2] \leq \mathcal{O}(n^{-r} + e^{(1-\frac{r}{2})}).$$

The estimation error converges to zero when $r \in (0, 2)$, and $r \to 2$ achieves a quadratic rate. $\square$

## A.3 ANALYSIS OF CONDITION IN THEOREM 1

We plot the trend of theorem holds in Figure 9 using three probability of semantic equivalence outputs of $\hat{\boldsymbol{y}}$, where $\psi \in \{0.1, 0.05, 0.03\}$. Each figure presents three minimal probability among all semantic equivalence outputs of $\hat{\boldsymbol{y}}$, denoted as $P_{min}$. The condition of Theorem 1 is satisfied when corresponding line is below the dashed line.

The results show that our proposed PC approach outperforms SC method when number of samplings $n$ is limited, which is confirmed by our experimental results. These results indicate that our proposed PC approach achieves better performance in cases where a larger optimal $n$ is required, such as when the sampling temperature is high, the problem being addressed is more complex, or there are more candidate answers to consider. The results also show that the proposed PC approach outperforms

the SC method when the number of samples n is sufficient. This is because, with a sufficiently large n, the PC approach is able to sample all possible outputs with a high probability, thereby achieving nearly zero variance due to the incorporation of token-level confidence.

Moreover, the conditions are related to the minimal probability among all semantic equivalence outputs of $\hat{y}$, which can be quite small. We argue that the minimal probability can be controlled by removing outputs with extremely small probabilities, as these low-probability outputs could also be incorrect. Our advanced version of the PC approach, i.e., RPC approach, has demonstrated that excluding such low-probability outputs can further improve overall performance.

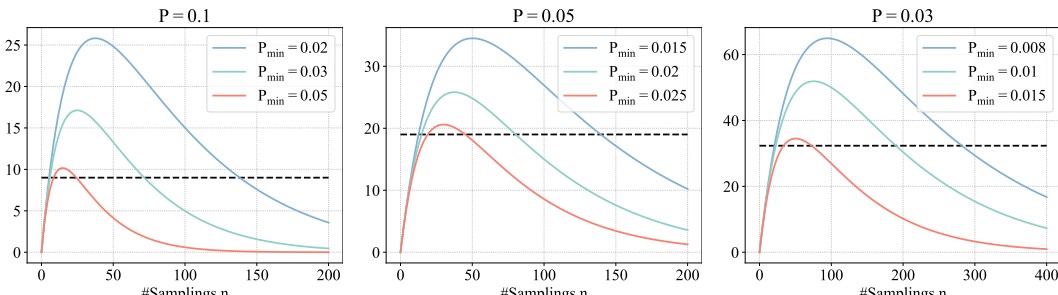

Figure 9: The trend indicated by the theorem holds, and the condition is satisfied when the line is below the dashed line

## B  DETAILS OF EXPERIMENTS SETTING

### B.1  DATASETS

For mathematical reasoning tasks, we evaluate our proposed methods and comparison methods on one common mathematical benchmark datasets, MATH as well as three challenging mathematical datasets that include Olympiad-level problems, i.e., MathOdyssey, OlympiadBench, and AIME datasets. MATH (Hendrycks et al., 2021b) is a dataset comprised of challenging competition math problems and we use its 5,000 testing data for evaluation. The MathOdyssey dataset (Fang et al., 2024) contains 387 problems, covering advanced high-school level, university-level, and Olympiad-level mathematics. The OlympiadBench dataset (He et al., 2024) contains 8,476 Olympiad-level mathematics and physics problems. We select the English problems without images, resulting in a testing dataset of 1,284 problems. The AIME dataset (Zamil & Rabby, 2024) contains 993 test problems collected from the American Invitational Mathematics Examination, spanning from 1983 to 2024.

For code generation tasks, we conduct experiments on three common benchmark datasets. HumanEval (Chen et al., 2021) contains 164 hand-written Python programming problems. MBPP (Austin et al., 2021)(sanitized version) consists of 427 entry-level programming problems. We also include the introductory-level problems of APPS (Hendrycks et al., 2021a), which contains 1000 problems.

### B.2  DETAILES OF CODE REASONING TASK

**Code generation.** On the code reasoning task, we let LLM generate a code snippet to solve a given programming problem, and then evaluate its functional correctness based on the ground-truth test cases provided by the dataset. In detail, we set the top $p$ to 0.95, the max generation length to 1024. For code snippet post-processing, we first extract the code text from code block surrounded by triple-backticks(```), and then we follow Chen et al. (2021) to truncate the generated code snippet before the following stop sequences: "\nclass", "\ndef", "\n#", "\nif", "\nprint". At the same time, we also obtain the log-probability of each token from the LLM response. For "verbalization" setting, the verbalized confidence is also extract from the text generated by LLM along with the code snippet.

**Self-consistency on code.** We follow Chen et al. (2022) to sample 100 test cases for each programming problem from the same model. And then we achieved self-consistency on code at the semantic equivalence level, which based on the execution behavior of any two codes on this set of

test cases. More formally, we implemented the consistency function $\mathbb{I}_C(\cdot, \cdot)$ as an indicator function that indicates whether two codes are semantically equivalent, i.e., $\mathbb{I}_C(x, y) = 1$ if and only if code $x$ and $y$ execute the same result on this set of test cases.

## B.3 PROMPT TEMPLATES

**Prompt for generating code.** The prompt for generating code consists of a header, a functional signature, and a docstring and LLM needs to implement the body of this function.

```python
from typing import List
def has_close_elements(numbers: List[float],
        threshold: float) -> bool:
""" Check if in given list of numbers,
are any two numbers closer to
each other than given threshold.
"""
```

```python
from typing import List
def has_close_elements(numbers: List[float],
        threshold: float) -> bool:
""" Check if in given list of numbers, are any two numbers closer to
each other than given threshold.
"""
pass
# check the correctness of has_close_elements
assert
```

Figure 10: An example of prompt for generating code in HumanEval dataset

**Prompt for generating test cases.** For generating test cases, we implemented the function body with a "pass" statement on the basis of the prompt to generate the code, and added a comment to require the LLM to generate test cases for the programming problem.

**Prompt for code verbalized method.** For generating code with verbalized confidence, we added instructions for generating verbalized confidence, as well as format requirements to facilitate the extraction of code and confidence score. We also gave a simple example to help LLM understand the format requirements at then end of the prompt.

**Prompt for math reasoning tasks.**

The InternLM2-MATH-Plus 1.8B and 7B models are chat models that facilitate conversations between two roles: "user" and "assistant". The prompt for the "user" role is provided in Prompt 1. Similarly, the prompt for the DeepSeek-Math 7B model is shown in Prompt 2.

**Prompt for math verbalized method.**

We observed that the tuned math models are challenging to prompt for generating confidence. Therefore, we adopted the methods from Tian et al. (2023) to calculate the probability based on the likelihood of the first generated "True" token and the first generated "False" token. The corresponding prompt is provided in Prompt 3.

へ

Come up with a solution that solves the following programming question and provide your confidence score in this solution like 0%, 10%, ... 100%.

```
import heapq as hq
def heap_queue_largest(nums,n):
'''
Write a function to find the n largest integers from a given list
of numbers, returned in descending order.
'''
```

Format requirement: output in the form of the following example. Do not provide any additional explanations. Here is an output example: Solution:

```python
your code ...
```

Confidence: ...

Figure 11: An example of prompt for verbalized confidence estimation in MBPP dataset.

---

**Prompt 1: Prompt for InternLM-2-Math-Plus**

Problem:\n{instruction}\n Let's think step by step\n Solution:\n

---

**Prompt 2: Prompt for DeepSeek-Math**

instruction\n Please reason step by step, and put your final answer within \\boxed{{}}.

---

**Prompt 3: Prompt for DeepSeek-Math**

Question: question\n Proposed Answer: answer\n Is the proposed answer:\n \t(A) True or\n \t(B) False?\n The proposed answer is:

## C  DETAILED EXPERIMENTAL RESULTS

### C.1  EXCLUDED AND RETAINED PERPLEXITY OF RPC APPROACH

We have plotted the excluded and retained confidence distributions using the *Reasoning Pruning* in the RPC approach across each dataset, with temperatures $T \in \{1.0, 1.1, 1.3\}$, in Figure 12. We believe this visualization will help researchers better understand the workings of our RPC, particularly how it filters out unreliable reasoning paths while retaining those with higher confidence.

### C.2  FULL EXPERIMENTAL RESULTS

In this paper, we conduct experiments using InternLM2-Math-Plus 1.8B model, InternLM2-Math-Plus 7B model, DeekSeek-Math 7B model under sampling temperature $T \in \{1.0, 1.1, 1.3\}$. The full experimental results measured by accuracy, ECE, and Brier score metrics are reported in Table 5, Table 4, Table 6. The results using the InternLM2-Math-Plus 1.8B model with $T = 1.3$ were excluded because the generated answers caused the answer checker process to be suspended.

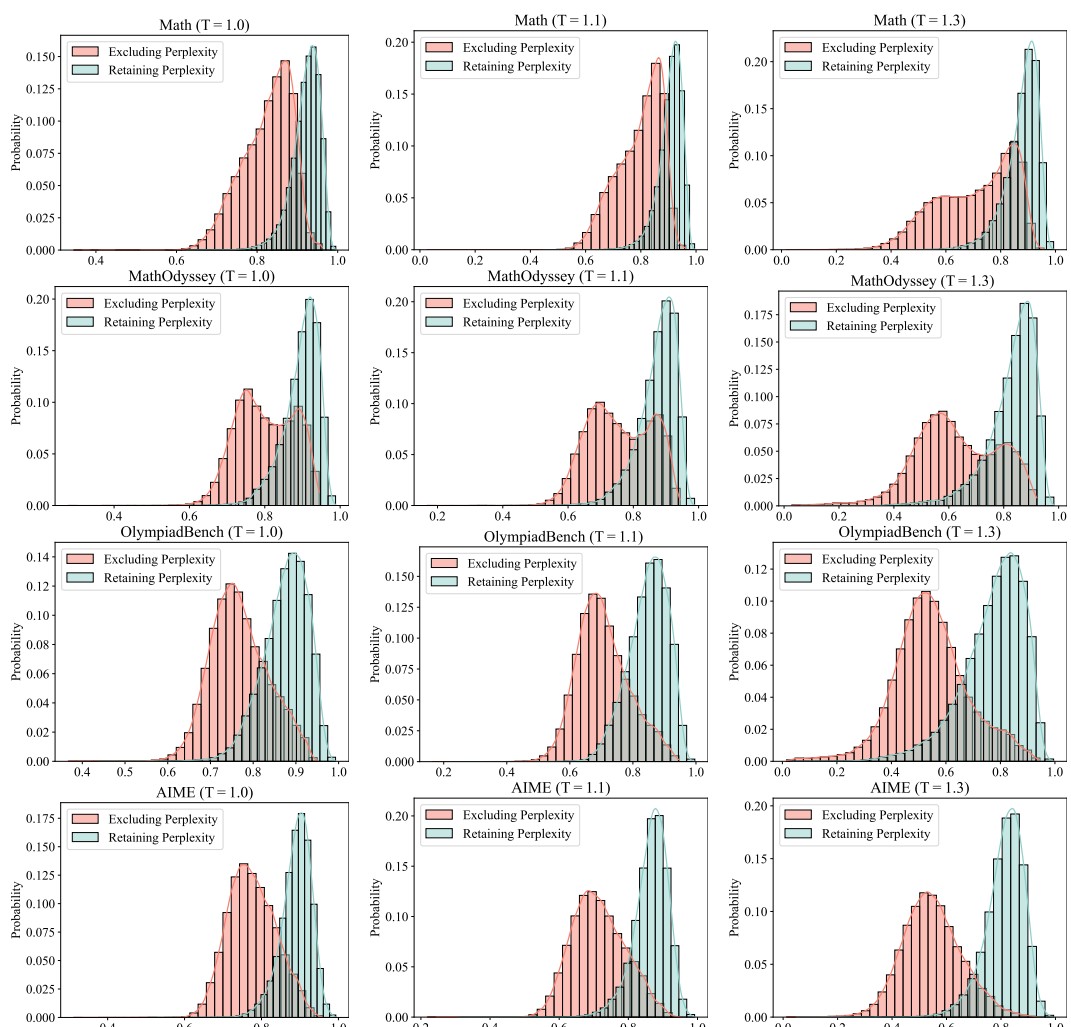

Figure 12: The excluded and retained confidence distributions of RPC approach.

We also plot the ECE diagrams for the PPL, SC, PC, and RPC methods under temperatures T ∈ 1.0, 1.1, 1.3 using the InternLM2-Math-Plus-7B model across four mathematical reasoning datasets, as shown in Figure 13, Figure 14, Figure 15, and Figure 16, respectively.

Table 4: Detailed results using InternLM-2-MATH-Plus 7B model. The best performance is highlighted in **bold**, while the second-best performance is underlined.

| Methods | MATH Accuracy (↑) | ECE (↓) | BS (↓) | MathOdyssey Accuracy (↑) | ECE (↓) | BS (↓) | OlympiadBench Accuracy (↑) | ECE (↓) | BS (↓) | AIME Accuracy (↑) | ECE (↓) | BS (↓) |
|---|---|---|---|---|---|---|---|---|---|---|---|---|
| | | | | | | Sampling Temperature $T=1.0$ | | | | | | |
| PPL | 46.68 | 49.52 | 49.13 | 25.45 | 70.04 | 67.81 | 7.17 | 87.60 | 83.25 | 5.14 | 90.30 | 86.35 |
| VERB | 25.81 | 48.12 | 43.51 | 9.37 | 71.14 | 64.02 | 3.49 | 85.77 | 79.38 | 3.13 | 87.52 | 80.40 |
| SC | 50.50 | 6.60 | **15.49** | 28.92 | 11.64 | 18.53 | 11.06 | 20.95 | 12.60 | 9.54 | 13.94 | **10.13** |
| PC | 50.86 | 6.30 | 15.69 | 28.28 | 12.08 | 18.18 | 11.06 | 20.71 | 12.46 | 9.65 | **13.79** | 10.18 |
| RPC | **52.32** | **6.29** | 15.68 | **33.16** | **8.59** | **18.11** | **11.21** | **19.45** | **11.88** | **9.75** | 13.94 | 10.32 |
| | | | | | | Sampling Temperature $T=1.1$ | | | | | | |
| PPL | 47.06 | 48.38 | 47.88 | 27.25 | 67.20 | 64.76 | 6.93 | 86.35 | 80.82 | 5.89 | 88.39 | 83.60 |
| VERB | 24.80 | 48.70 | 43.82 | 9.32 | 73.07 | 66.12 | 3.88 | 85.61 | 79.09 | 3.17 | 87.27 | 80.10 |
| SC | 50.79 | 5.04 | **15.43** | 28.41 | 10.82 | 18.28 | 10.79 | 20.18 | 12.06 | 8.57 | 13.61 | **9.08** |
| PC | 50.98 | 4.94 | 15.49 | 28.79 | 10.27 | 18.23 | 10.83 | 19.77 | 11.90 | 8.68 | 13.47 | 9.13 |
| RPC | **53.18** | **4.66** | 15.83 | **34.19** | **6.13** | **17.95** | **11.14** | **18.28** | **11.21** | **9.75** | **12.74** | 10.20 |
| | | | | | | Sampling Temperature $T=1.3$ | | | | | | |
| PPL | 47.90 | 46.21 | 45.73 | 25.45 | 67.14 | 63.66 | 7.55 | 82.90 | 75.36 | 6.65 | 85.22 | 78.61 |
| VERB | 23.89 | 49.42 | 44.32 | 8.39 | 76.13 | 67.61 | 3.23 | 86.18 | 79.34 | 2.27 | 88.36 | 80.97 |
| SC | 50.70 | **2.22** | **15.02** | 27.63 | 10.36 | 17.05 | 10.40 | 17.82 | 10.53 | 8.20 | 11.62 | **8.29** |
| PC | 51.42 | 2.26 | 15.20 | 28.02 | 10.03 | 16.83 | 10.67 | 17.07 | 10.41 | 8.68 | **10.99** | 8.68 |
| RPC | **53.44** | 2.67 | 15.41 | **32.13** | **5.75** | **16.72** | **11.29** | **15.63** | **10.28** | **8.90** | 11.00 | 8.99 |

Table 5: Detailed results using InternLM2-MATH-Plus 1.8B model. The best performance is highlighted in **bold**, while the second-best performance is underlined.

| Methods | MATH Accuracy (↑) | ECE (↓) | BS (↓) | MathOdyssey Accuracy (↑) | ECE (↓) | BS (↓) | OlympiadBench Accuracy (↑) | ECE (↓) | BS (↓) | AIME Accuracy (↑) | ECE (↓) | BS (↓) |
|---|---|---|---|---|---|---|---|---|---|---|---|---|
| | | | | | | Sampling Temperature $T=1.0$ | | | | | | |
| PPL | 32.44 | 62.70 | 60.90 | 15.94 | 78.61 | 75.22 | 2.26 | 91.58 | 86.11 | 1.39 | 93.49 | 88.81 |
| VERB | 6.40 | 41.54 | 38.39 | 2.27 | 61.91 | 50.05 | 0.58 | 74.32 | 58.58 | 0.20 | 77.03 | 61.05 |
| SC | 36.61 | 6.28 | **15.04** | 14.40 | 18.73 | 14.61 | 6.07 | 21.77 | 11.37 | 2.68 | 16.60 | 6.38 |
| PC | 36.88 | **6.11** | 15.29 | 14.65 | 18.34 | 14.55 | 6.07 | 21.54 | 11.20 | 2.68 | 16.49 | **6.33** |
| RPC | **38.16** | 6.14 | 15.45 | **15.68** | **16.78** | **14.23** | **6.54** | **20.23** | **10.78** | **3.43** | **15.82** | 7.03 |
| | | | | | | Sampling Temperature $T=1.1$ | | | | | | |
| PPL | 32.90 | 61.32 | 59.19 | 16.45 | 76.83 | 72.60 | 2.80 | 89.09 | 82.15 | 2.14 | 91.28 | 85.49 |
| VERB | 6.00 | 44.00 | 39.55 | 2.25 | 64.42 | 52.11 | 0.33 | 78.47 | 64.38 | 0.14 | 78.88 | 63.60 |
| SC | 36.77 | 4.19 | **14.82** | 12.98 | 18.55 | **13.75** | 5.69 | 20.52 | 10.32 | 2.04 | 16.42 | 5.54 |
| PC | 36.94 | 3.97 | 14.85 | 13.62 | 17.78 | 13.90 | 5.76 | 20.28 | 10.33 | 2.14 | 16.09 | **5.54** |
| RPC | **38.66** | **3.87** | 15.11 | **16.20** | **15.32** | 14.49 | **6.15** | **19.61** | **10.23** | **2.68** | **15.34** | 5.93 |

Table 6: Detailed results using DeekSeek-Math 7B model. The best performance is highlighted in **bold**, while the second-best performance is underlined.

| Methods | MATH Accuracy (↑) | ECE (↓) | BS (↓) | MathOdyssey Accuracy (↑) | ECE (↓) | BS (↓) | OlympiadBench Accuracy (↑) | ECE (↓) | BS (↓) | AIME Accuracy (↑) | ECE (↓) | BS (↓) |
|---|---|---|---|---|---|---|---|---|---|---|---|---|
| | | | | | | Sampling Temperature $T=1.0$ | | | | | | |
| PPL | 41.52 | 55.00 | 54.81 | 21.59 | 74.94 | 73.50 | 5.22 | 91.31 | 88.35 | 2.89 | 94.22 | 91.60 |
| VERB | 13.02 | 58.53 | 56.33 | 1.76 | 91.67 | 90.34 | 2.09 | 91.19 | 89.51 | 1.70 | 92.82 | 90.59 |
| SC | 53.54 | **6.11** | 16.53 | 36.25 | 11.07 | 17.56 | 11.49 | 15.24 | 10.67 | 9.36 | 11.91 | **9.06** |
| PC | 53.56 | 6.23 | 16.64 | 36.25 | 10.78 | **17.41** | 11.45 | **15.18** | **10.58** | 9.65 | 11.51 | 9.34 |
| RPC | **53.58** | 6.26 | 16.69 | **37.28** | **9.55** | 17.78 | **11.60** | 15.21 | 10.77 | **9.86** | 11.52 | 9.26 |
| | | | | | | Sampling Temperature $T=1.1$ | | | | | | |
| PPL | 42.42 | 53.56 | 53.42 | 23.14 | 72.89 | 71.43 | 5.92 | 89.99 | 86.56 | 3.43 | 93.09 | 89.99 |
| VERB | 12.57 | 59.40 | 57.04 | 1.91 | 91.67 | 89.85 | 2.01 | 92.24 | 90.32 | 1.77 | 91.97 | 89.77 |
| SC | 53.78 | **4.97** | **16.21** | 37.28 | **8.94** | 17.52 | 11.57 | 14.44 | 10.42 | 9.06 | 11.78 | **8.78** |
| PC | 53.80 | 4.98 | 16.39 | **37.28** | 9.05 | 17.45 | **11.60** | **14.34** | 10.39 | 9.22 | 11.53 | 8.88 |
| RPC | **53.80** | 5.14 | 16.46 | 37.02 | 9.73 | **17.22** | 11.45 | 14.86 | **10.26** | **9.65** | **11.14** | 8.99 |
| | | | | | | Sampling Temperature $T=1.3$ | | | | | | |
| PPL | 43.62 | 51.23 | 51.14 | 25.19 | 69.41 | 67.59 | 6.54 | 87.97 | 83.51 | 3.86 | 91.37 | 87.22 |
| VERB | 12.12 | 60.18 | 57.51 | 1.75 | 91.79 | 90.24 | 1.65 | 92.61 | 90.90 | 1.54 | 93.25 | 91.08 |
| SC | 54.29 | **3.50** | 15.97 | 38.43 | **7.66** | **17.66** | 10.98 | 13.42 | 9.19 | 9.49 | **9.89** | 8.74 |
| PC | 54.38 | 3.55 | 16.16 | 38.82 | 7.80 | 17.72 | 10.90 | 13.47 | **9.13** | 9.43 | 9.96 | 8.71 |
| RPC | **54.54** | 4.22 | 16.18 | **39.07** | 7.90 | 17.70 | **11.45** | 13.52 | 9.53 | **9.75** | 10.42 | **8.66** |

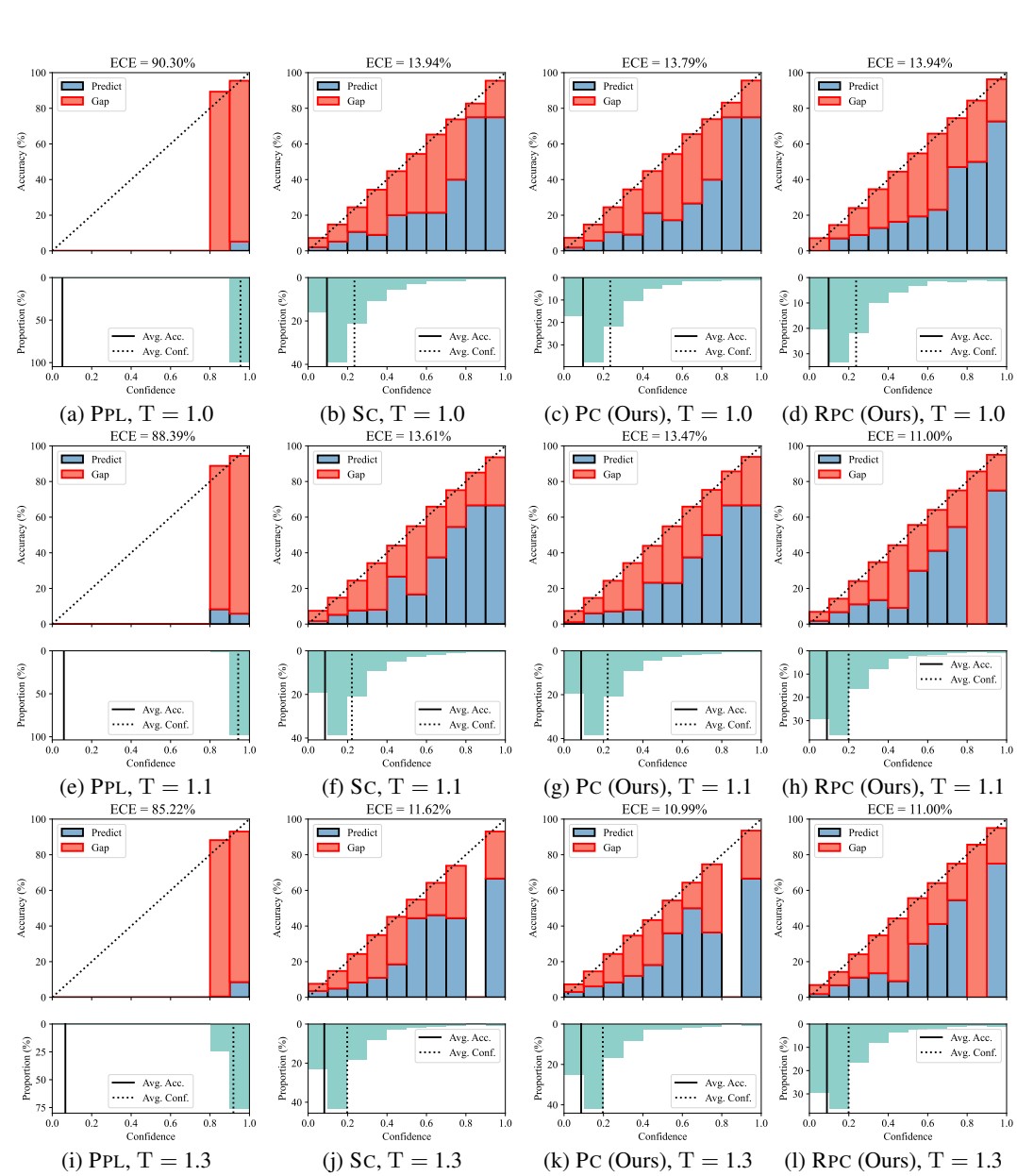

Figure 13: The ECE of InternLM-2-MATH-Plus 7B model on AIME datasets with temperatures $T \in \{1.0, 1.1, 1.3\}$ and sample size $n = 256$.

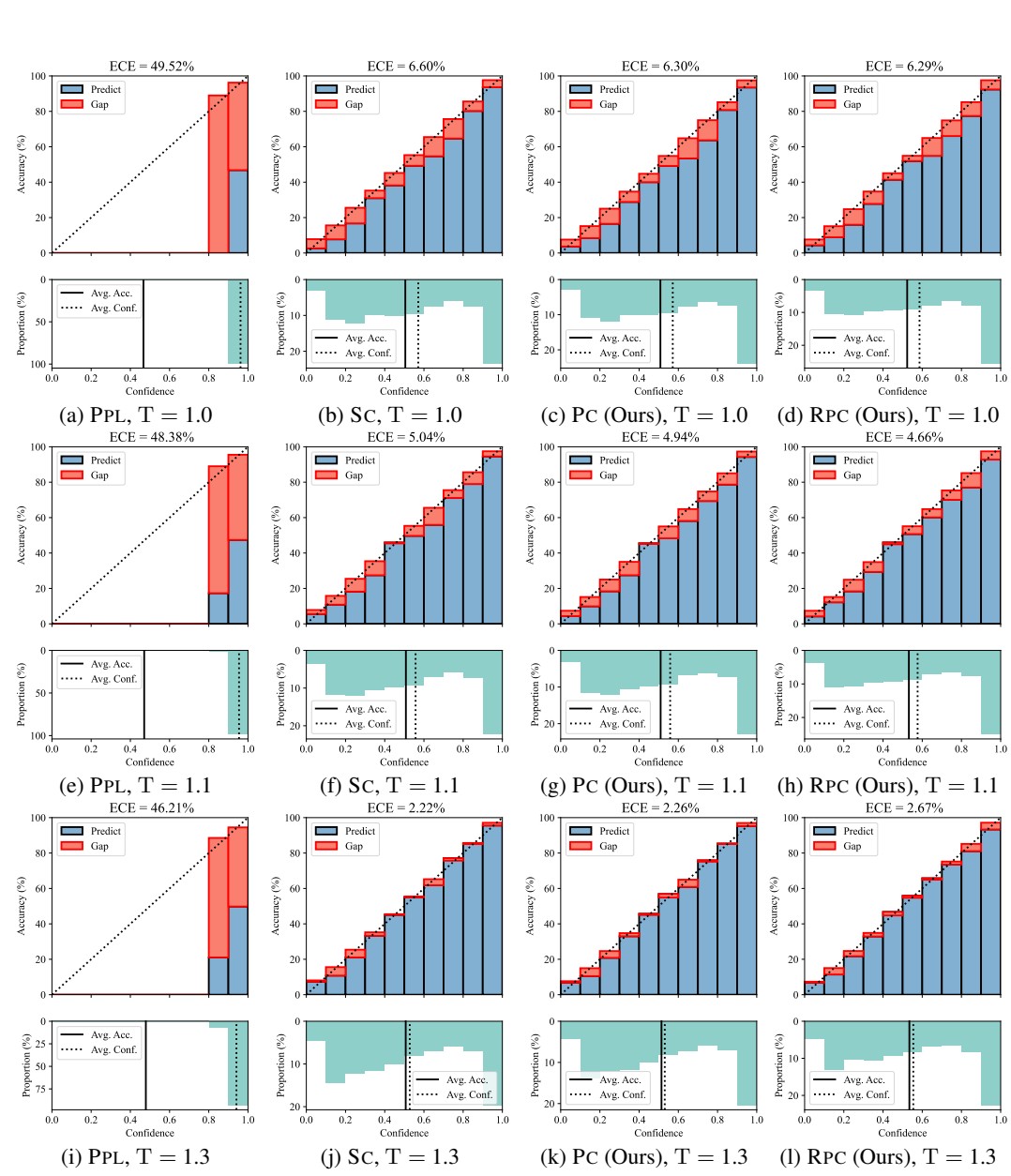

Figure 14: The ECE of InternLM-2-MATH-Plus 7B model on MATH datasets with temperatures $T \in \{1.0, 1.1, 1.3\}$ and sample size $n = 100$.

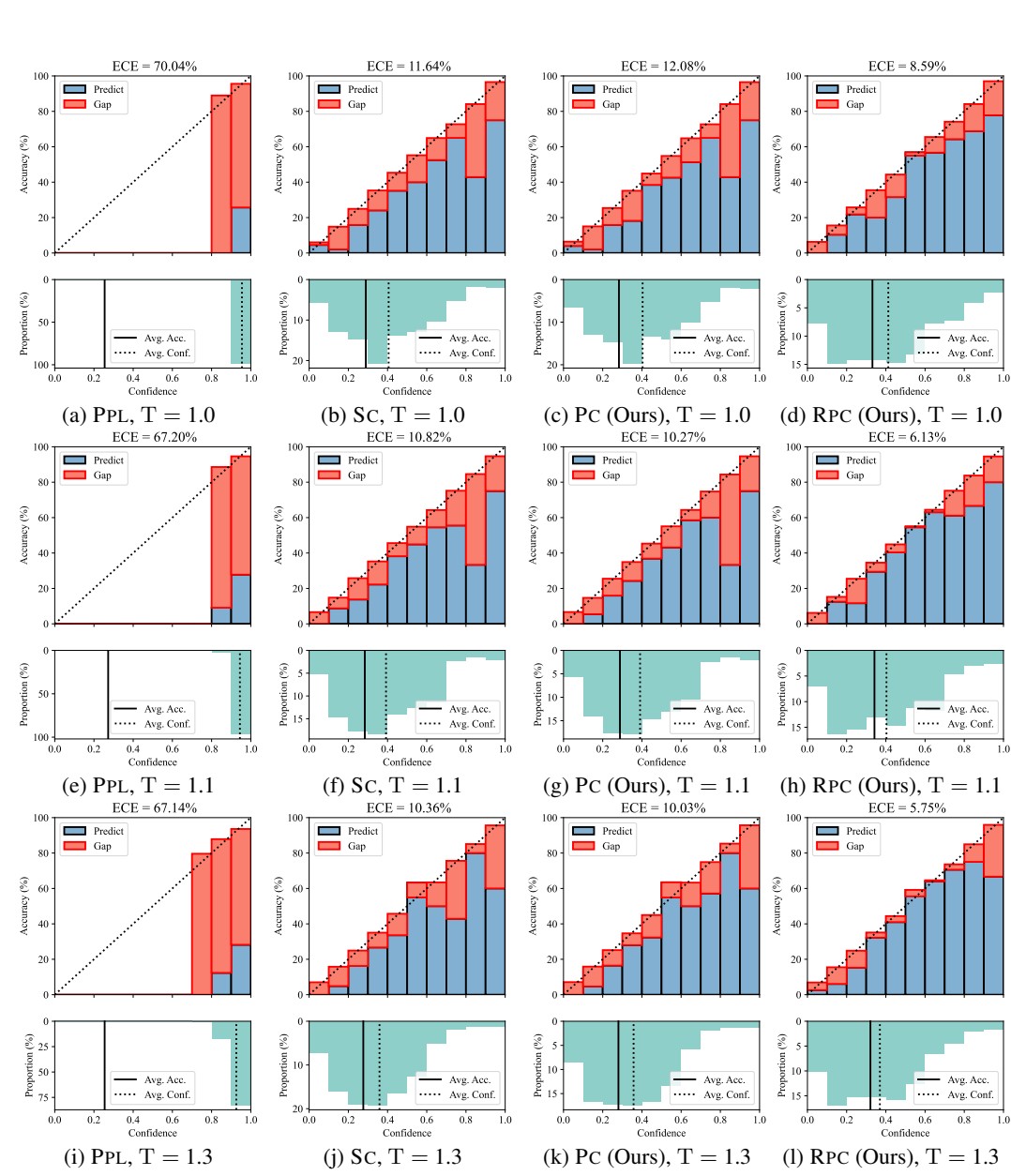

Figure 15: The ECE of InternLM-2-MATH-Plus 7B model on Odyssey datasets with temperatures $T \in \{1.0, 1.1, 1.3\}$ and sample size $n = 256$.

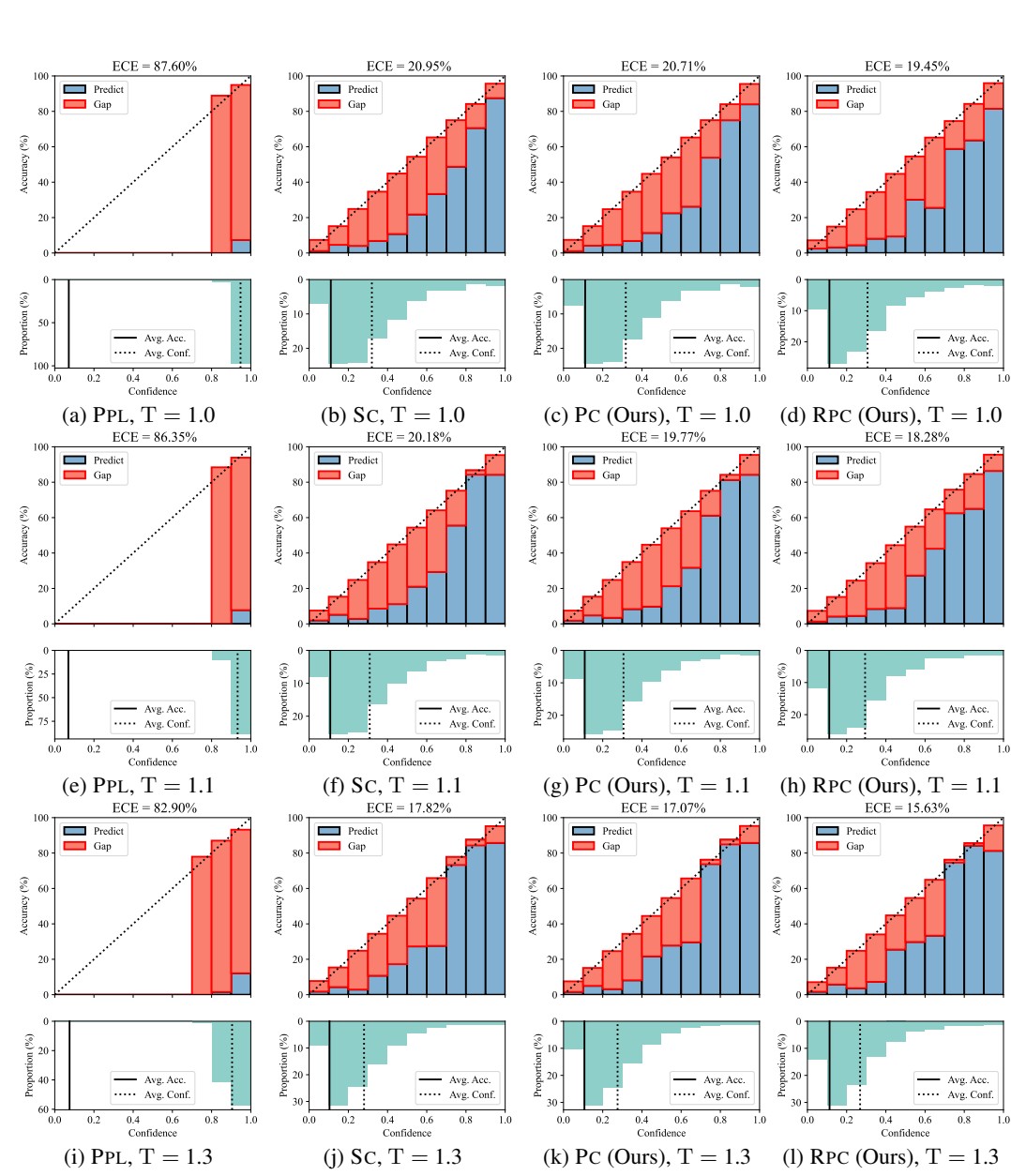

Figure 16: The ECE of InternLM-2-MATH-Plus 7B model on OlympiadBench datasets with temperatures T ∈ {1.0, 1.1, 1.3} and sample size $n = 256$.

