# OpenReview forum: "Towards Efficient Confidence Estimation for Large Language Model Reasoning"
_ICLR.cc/2025/Conference — ICLR 2025 Conference Withdrawn Submission_

### Official Review · Reviewer_KNWA · 2024-11-04

**Soundness:** 2
**Presentation:** 3
**Contribution:** 3
**Rating:** 5
**Confidence:** 3

**Summary:**

This paper introduces two methods, Perplexity Consistency (PC) and Reasoning-pruning Perplexity Consistency (RPC), for confidence estimation in LLM reasoning under resource constraints. The proposed methods integrate prediction probabilities (for multiple reasoning chains) and prune low-probability reasoning chains to reduce variance. Theoretical analyses support their approach, and experiments across seven benchmark datasets demonstrate that PC and RPC outperform existing confidence estimation methods in accuracy and generally in calibration.

**Strengths:**

1. **Theoretical Motivation:** The paper is well supported by theoretical arguments, along with relevant experiments.
2.  **Important Contribution:** The paper studies an important problem of confidence estimation in LLMs (in a reasonable resource-constraint setting) along with improving their accuracy when sampling multiple reasoning paths.
3. **Comprehensive Evaluations:** The evaluations are comprehensive in terms of the number of relevant datasets and models.

**Weaknesses:**

1. The results on ECE are not particularly better often offering little or limited advantage compared to self-consistency. Further, lack of statistical significance experiments doesn't make it clear if the method can consistently provide better calibration than baselines.
2. Additionally, it isn't clear for what difficulty of question, does the method actually provide better performance. For instance, does SC perform worse than RPC in Figure 6 just because of the second last bin? Also, why is there so much difference in that particular bin? (similar cases are visible in the appendix)
3. The method was not evaluated on lower temperatures, (eg, 0.3, 0.7) which are fairly standard settings in practice. Is there a particular limitation of the method for lower temperatures?
4. It would strengthen the paper if certain base-models (without RLHF and instruction tuning) models were also evaluated since they are usually better calibrated.
5. The RPC method introduces several new parameters that need to be learned on a training dataset. This introduces additional overhead. Can authors provide details for the same?

**Questions:**

See the Weaknesses above.

Additionally,

1. Can authors share some qualitative examples (in terms of tokens probability along with generated answers) to demonstrate how and for which kind of queries their method peak's accuracy is better than self-consistency?
2.  Do authors have an intuition of significantly better peak performance in the MathOdyssey dataset compared to other datasets such as AIME and OlympiadBench?

---

### Official Review · Reviewer_CZeH · 2024-11-04

**Soundness:** 2
**Presentation:** 1
**Contribution:** 2
**Rating:** 3
**Confidence:** 3

**Summary:**

The paper addresses the problem of improving confidence estimation for LLMs, aiming to use these estimates to enhance performance. This is achieved by removing certain tokens from each step in the generation process based on their confidence estimates. Importantly, the confidence estimates leverage the outcome probabilities of the LLM. The authors propose a novel adaptation for self-consistency, which typically suffers from the need for too many samples for calibration, by utilizing the predicted probabilities instead of relying on Monte Carlo-based sampling estimations. They evaluate their technique on several benchmarks, demonstrating an improvement of approximately 2-5% over baselines across different domains, including code generation and question answering.

**Strengths:**

- The paper investigates a critical aspect of  large language models.

**Weaknesses:**

- Marginal improvement of the proposed solution as showcased in the reported results

- The paper clarity is very low, which requires a further proofreading  not only to fix grammatical and structural issues but improve the quality of the text


-   Methodology:
        -   In the methodology, the authors loosely define $\mathbb{I}_T$. This makes it harder to follow the presented derivations, and a clear presentation of the definition would go a long way.
        -   In Figure 4, when the improvement of the proposed technique is shown over baselines: the improvement (less than 4\%) seems to only happen when $n$ is greater than 30. This brings forth two questions: (i) is this still consistent with the observations made in the motivational section? And (ii) since the improvement is small, is the proposed technique worthwhile?
        -   More than once, claims about improved complexity are made. This doesn't seem to be measured in the experiments, and it would be nice to see figures/numbers to corroborate the theoretical results.
    -   Terminology:
        -   "Reasoning path" seems to be loosely defined as the ordered output tokens. This might be valid for the given domain of this work, but does not really align with works in mechanistic interpretability. Perhaps a clear statement of this, early on, would help set the stage properly for this work.
        -   The work seems to intervene on which tokens are allowed in the generation process, and this is only mentioned at the end of the methodology section. This reviewer believes that this should be stated more clearly in the beginning, to position the work properly. Keeping this in mind also makes it easier to read the experimental evaluation section, because the accuracy can then be understood as number of correct solutions (out of the total) given this modification of the generation process.


Overall, I think that the contribution of this paper can be made more clear by improving the presentation and clarifying certain details. Further, since this seems to be a highly empirical work, I think that it would benefit from a deeper statistical analysis to show the advantage of using it, i.e. to directly address the basic questions of "is a 2\% improvement relevant?" (or can it be attributed to random chance).

**Questions:**

-    The claim "accurate LLM probabilities" is repeated several times in the paper. It is unclear what the authors formally define as accurate probability?


-   In the beginning of Appendix A, a claim is made that $\mathbb{I}_C$ has a Bernoulli distribution. This is a non-trivial claim that needs to be justified, especially since the authors mention that the Jaccard similarity (i.e. non-binary) can be used to calculate $\mathbb{I}_C$. A concrete definition of $\mathbb{I}_C$ might be sufficient to answer this concern

-   General Comments:
        -  The standard autoregressive definition for LLM probabilities given in Line 135 (Eq. 4) => the final term should be for t < m, instead of m-1 ? Same remark in Line 143 (Eq. 5)


        -   Several comments are made regarding "small sample size," even though no actual number or bound is given to indicate this. The authors provide one statistic, namely that 100 samples for each of 5,000 problems would require 18 GPU hours. The figures provided (Figure 2 and 4) show that it is likely that half of this number of samples is more than sufficient. I believe adding a statement on what constitutes a sufficient sample size would go a long way

 -   The entirety of the work seems to ride on the quality of the probabilities output by the model, which begs the question: how does this perform when it comes to hallucination? This isn't to say that the focus should be moved to hallucination, but instead: how does the specific technique perform (i.e. what are the confidence estimates) when the model hallucinates?

-   Motivational Analysis:
        -   If the point behind Figure 2.a. is to show that convergence takes longer on complex tasks, why are more samples not shown? The AIME curve just keeps increasing, but does not seem to really converge for the displayed values of n. A simple fix of this would be to show the curve for more values of n.
        -   Figure 2.c. shows that higher temperatures (1.1 and 1.3) improve performance over low temparatures (0.3 and 0.5). However, this improvement seems to be very small (less than 2\%) beyond n=1, which begs the question: is the improvement really worthwhile?

---

### Official Review · Reviewer_Uy5W · 2024-11-11

**Soundness:** 1
**Presentation:** 3
**Contribution:** 2
**Rating:** 3
**Confidence:** 4

**Summary:**

This paper studies the problem of uncertainty estimation in LLMs. While there are many existing approaches for uncertainty estimation, the paper claims that they suffer from slow convergence and require many samples to achieve good performance. To solve the problem, this paper proposes a new uncertainty measure that combines local and global consistency measures. Specifically, given a problem (as prompts) x, the model samples multiple y’s. For each y, the proposed method computes both a token-level consistency and a global consistency. The final estimator is the average (over sampled y’s) of the product of the two estimates.

The paper provides theoretical guarantees on the variance of the estimator and demonstrates improved uncertainty estimation performance on several reasoning benchmarks.

**Strengths:**

- This paper studies the uncertainty estimation problem in LLMs, which is a very important problem for reliable LLM generation.
- The paper is generally well-written and has included sufficient background knowledge for a smooth reading.
- Empirical results show that the proposed technique can improve LLMs’ reasoning performance in many benchmarks.

**Weaknesses:**

- The paper should make a clear distinction between “confidence”/uncertainty and performance. While the primary objective of this paper is to improve uncertainty estimation, many results are actually about improving the accuracy of many reasoning tasks, such as Figure 2 (a, c), Figure 4, and Table 2. I found it a bit unclear between improve uncertainty estimation and improving performance — when improving performance we are implicitly “sampling” from a “better” (e.g., temperature scaled) distribution, but uncertainty/confidence estimation cares more about the “confidence” of the *original predictions*. It seems from the results that the proposed confidence estimation algorithm is good at both confidence estimation and improving performance. Could the author please clarify this?
- I am concerned about the statements made in the theorems. Theorem 1 states “With a proper assumption, ….” but the assumption is not mentioned in the main text, making it impossible to understand the relevance of the theorem. Theorem 2 does not seem to hold if $p_{\theta}^{(TL)}$ is chosen to be the log-likelihood and the distribution of y given x is uniform. In this case the two estimates seem to have the same rate. Maybe there are necessary assumptions missing from the statement.

**Questions:**

- While the paper claims that the proposed confidence estimation objective converges faster, is it more well-calibrated for certain distributions?
- How does the accuracies compare to approaches such as greedy temperature scaling?

---

### Official Review · Reviewer_Fnts · 2024-11-11

**Soundness:** 3
**Presentation:** 3
**Contribution:** 3
**Rating:** 6
**Confidence:** 4

**Summary:**

This paper tries to address the problem of estimating LLM reasoning confidence with a limited sample size and no additional calibration set (thus no temperature tuning). Their presented method is to integrate the prediction probability of LLMs into the self consistency confidence estimation, which they refer to as PC approach. They combine PC with Reasoning Pruning (to model the confidence distribution and automatically remove the reasoning paths with low probability) and sees improved performance across 7 benchmarks.

**Strengths:**

- The presentation of this paper is clear with a natural and logical flow. I especially appreciate how the authors include necessary (and only the necessary preliminaries before the main body of this paper). The problem setup and method description were also well structured and soundly put. The authors motivate the need of this research and the establishment of the main RQ well (e.g. the Challenge paragraph in Sec. 3) which I find convincing.

- The experiments are well designed and conducted across multiple important datasets/benchmarks in both math and coding with various SoTA models.

- The results are well discussed, and the generalization study from math to coding sounds natural and important. I appreciate the authors' attempt to propose a method that can be generalized across domains.

**Weaknesses:**

- I agree that math and coding are probably among the most important forms of reasoning, while I would still be curious to see the results of applying the proposed method on other reasoning types, e.g. reasoning tasks in natural languages such as commonsense reasoning. However, I don't want to post this point in the `Questions` section because to investigate this might be too much work to do during rebuttal. Don't worry too much about this -- the results here already look promising to me.

- Since this work has been experimenting with math and coding reasoning, I think it is natural to discuss a highly relevant field -- formal mathematical reasoning (e.g. in Lean), which is a natural combination and mathematical reasoning and code generation. I wonder how similar methods may be of help in formal math reasoning. I think this is a rather insufficiently explored area in AI for formal math.

**Questions:**

- In the experiment setup in the beginning of Sec. 3, the authors present the use of the MATH dataset (which was introduced in 2021) and the InternLM-2-MATH-Plus 7B (which was released in 2024). I wonder how the effects of potential data contamination are assessed/discussed/eliminated.

- Similarly, I am curious about how data contamination comes into play for the datasets chosen in Sec. 5 (main experiment).

---

### Note · Authors · 2024-11-20

**Comment:**

Dear Reviewers,

We would like to express our sincere thanks for your time and thoughtful feedback on our paper. We greatly appreciate the insightful comments you have provided. After carefully considering your points, we agree with most of the concerns raised. In particular:

* We acknowledge that the clarity of the paper can be improved. Specifically, we plan to expand our discussion on uncertainty estimation and performance improvement. Additionally, we will revise the statements of theorems to enhance their clarity.

* We recognize the need for more evidence to support the generalization of the proposed solution. To address this, we will conduct additional experiments on other reasoning tasks (e.g., commonsense reasoning), incorporate more baseline methods (e.g., greedy temperature scaling), and test additional base models (e.g., RLHF and instruction-tuned models).

* We will provide more detailed analysis in the experiments section, including a more thorough examination of data contamination, statistical significance, and the additional overhead introduced by our proposed method.

Given the limited rebuttal period, which does not allow sufficient time to fully address these points, we have decided to withdraw the paper at this time.

Thank you once again for your valuable feedback.

Best,

The Authors

**Withdrawal Confirmation:**

I have read and agree with the venue's withdrawal policy on behalf of myself and my co-authors.